# Synthesis and Biological Evaluation of Small Molecules as Potential Anticancer Multitarget Agents

**DOI:** 10.3390/ijms23137049

**Published:** 2022-06-24

**Authors:** Alberto Pla-López, Raquel Castillo, Rocío Cejudo-Marín, Olaya García-Pedrero, Mariam Bakir-Laso, Eva Falomir, Miguel Carda

**Affiliations:** 1Inorganic and Organic Chemistry Department, Universidad Jaume I, 12071 Castellón, Spain; apla@uji.es; 2Physical and Analytical Chemistry Department. Universidad Jaume I, 12071 Castellón, Spain; rcastill@uji.es; 3Predepartmental Medicine Unit, Universidad Jaume I, 12071 Castellón, Spain; rcejudo@uji.es; 4Instituto de Química Organometálica Enrique Moles, Centro de Innovación en Química Avanzada, Universidad de Oviedo, 33006 Oviedo, Spain; al386307@uji.es; 5Instituto Universitario de Investigación en Ciencias Ambientales de Aragón (IUCA), Universidad de Zaragoza, 50009 Zaragoza, Spain; mbakir@unizar.es

**Keywords:** PD-L1, VEGFR-2, c-Myc, multitarget inhibitors, immunomodulation, angiogenesis, non-peptidic small molecules, flow cytometry

## Abstract

Twenty-six triazole-based derivatives were designed for targeting both PD-L1 (programmed death receptor ligand 1) and VEGFR-2 (vascular endothelial growth factor receptor 2). These compounds were synthetized and biologically evaluated as multitarget inhibitors of VEGFR-2, PD-L1 and c-Myc proteins. The antiproliferative activity of these molecules on several tumor cell lines (HT-29, A-549, and MCF-7) and on the non-tumor cell line HEK-293 was determined. The effects on the abovementioned biological targets were evaluated for some selected compounds. Compound **23**, bearing a *p*-chlorophenyl group, showed better results than sorafenib in regard to the downregulation of VEGFR-2 and a similar effect to BMS-8 on both PD-L1 and c-Myc proteins. The antiangiogenic and antivascular activities of chloro derivatives were also established by endothelial microtube formation assay on Matrigel^®^.

## 1. Introduction

In 2001, the US Food and Drug Administration (FDA) approved the tyrosine kinase inhibitor imatinib mesylate (Gleevec^®^) for the treatment of chronic myeloid leukemia. The introduction of this drug in clinical oncology was a great breakthrough in the combat against cancer. Somehow this drug can be considered as the starting point of a new modality in cancer treatment known as targeted therapy, which encompasses treatments that use non-peptidic small molecules, monoclonal antibodies, cancer vaccines and gene therapy [1]. In targeted therapy, drugs inhibit specific genes and proteins that are involved in the growth and spread of cancer cells, rather than by acting on rapidly dividing cells. Since imatinib’s approval, a relatively large number of new non-peptidic small molecules have been approved for their use in oncological treatments. Targeted therapy drugs first focused their action on the inhibition of certain proteins or cell signaling pathways such as receptor and non-receptor tyrosine kinases and serine/theonine kinases. Later, the field of action of this class of drugs was extended to other biological targets such as proteasome, proteins related to the Hedgehog pathway or epigenetic activity; proteins of the BCL-2 family that regulate the intrinsic apoptosis pathway; and poly(ADP-ribose)polymerases (PARPs), a group of enzymes that are engaged in DNA repair [2]. In contrast to large biomolecules, such as antibodies, non-peptidic small molecules offer some advantages, as they can target not only the extracellular components, such as cell surface receptors or glycoproteins attached to the cell membranes, but they can also reach intracellular proteins, as they are capable to easily cross the outer plasma membrane of the cell. In addition, these non-peptidic small molecules are cheaper than antibodies and many of them can be administered to the patient orally.

In the last decade, some oncological treatments were focused on the use of the immune system to fight cancer, so that immunotherapy has emerged as a major therapeutic modality in oncology [3] and novel biological targets, such as immune checkpoint proteins to prevent autoimmunity, have achieved a great relevance. Programmed cell death protein 1 (PD-1, CD-279) and programmed death 1 ligand (PDL-1, CD-274) play a crucial role in boycotting the immune response. In the last years, six antibodies that target these checkpoint receptors have been approved by regulatory agencies. However, the development of immunomodulatory non-peptidic small molecules lags behind the development of antibodies, even though these small molecules are expected to regulate intracellular signaling downstream of checkpoint proteins in both immune and cancer cells [4].

PD-L1 is overexpressed on cancer cells and induces immune tolerance. Together with a multitude of other proteins, these checkpoint receptors constitute molecular elements of the immunological system. Therefore, small-molecule immunotherapy can provide an alternative treatment modality either alone or in association with extracellular checkpoint monoclonal antibodies (mAbs) in the cases of low clinical responses or drug resistances [5].

In 2015, Brystol-Myers Squibb reported the first non-peptidic small molecules that are able to interact with PD-L1. In 2016, the mode of interaction in the PD-L1 system of some of these compounds, named BMS-202 and BMS-8 (see Figure 1), was established [6].

On the other hand, c-Myc is a transcription factor that, when overexpressed in cancer cells, induces PD-L1 overproduction and becomes responsible for the prevention of immune cells from attacking tumors. It has been established that c-Myc plays a fundamental role in building an immunosuppressive tumor microenvironment through the recruitment of tumor-associated macrophages or through the upregulation of the checkpoint proteins CD47 and PD-L1 [7,8]. Moreover, c-Myc is also responsible for the promotion of the overproduction of endothelial growth factor VEGF and its receptor VEGFR-2 in both cancer and endothelial cells. It has already been established that VEGF and VEGFR-2 are overexpressed in subsets of tumor cells, thus contributing to tumorigenesis, in addition to angiogenesis, through an autocrine mode of action [9,10]. Many non-peptidic small molecules targeting VEGFR have been approved by the FDA. Figure 2 depicts the structures of some of these inhibitors, which can be considered as multi-kinase inhibitors since, apart from VEGFR, they are capable of inhibiting many other kinases.

The 1,2,3-triazole ring is a five-membered heterocycle, and some compounds bearing this functionality, such as antibacterial Tazobactum and antibiotic Cefatrizine, have reached clinical practice [11]. The triazole ring is endowed with features such as hydrogen bond formation and ion–dipole, dipole–dipole, and π-stacking interactions that could explain its wide impact in the field of medicinal chemistry [12].

Our goal in this study was the development of small molecules bearing a 1,2,3-triazole central ring that could target both PD-L1 and VEGFR-2 in tumor cells. For 1,2,3-triazole ring-containing structures that have been shown to be active on VEGFR2 inhibition, see References [13,14,15,16,17,18,19]. We were also interested in checking the effect of these molecules on c-Myc. First, by docking studies we rationally designed some scaffolds that could fit PD-L1 protein so that they could act as inhibitor agents. Then we virtually checked if these scaffolds could also fit to VEGFR-2 in order to generate multitarget agents [20,21].

## 2. Results

### 2.1. Docking Studies

To start with, we decided to identify possible small molecules that could inhibit both PD-L1 and VEGFR-2 proteins by carrying out docking studies, using AutoDock Vina software [22]. We designed different ligands by modifying the inhibitor BMS-202 and the potent VEGFR-2 inhibitor sorafenib [23] using the Molden program [24].

Based on our previous studies about designing multitarget inhibitors [20,21], we obtained the starting coordinates from the RCSB Protein Data Bank with PDB ID 5J89 for PD-L1 system and 4ASD for VEGFR-2 system. One of the possible structures that fits well in both systems is the simple triazole system, which is shown below in Figure 1 (R_1_ = H, R_2_ = H). Figure 3a shows the superposition of simple triazole **1** (red) and BMS-202 (gray) at the PD-L1 binding site. Figure 3b shows the superposition of simple triazole **1** (red) and sorafenib (gray) at the tyrosine kinase domain in VEGFR-2. Thus, triazole **1** is located in a similar position as the BMS-202 and sorafenib. In the PD-L1 system, both the aromatic rings of compound **1** are interacting with Tyr56 of chain A (T stacking) and Tyr56 of chain B (π stacking), while in VEGFR-2 (see Figure 3b) triazole **1** occupies the hydrophobic cavity formed by Ala866, Val899, Phe918, and Leu1035.

We also studied the *p*-aniline triazole derivative with R_1_ = NH_2_ (compound **14** in Figure 1). Figure 3c,d shows that compound **14** fits both PD-L1 and VEGFR-2, with conformations similar to BMS-202 in PDL-1 and sorafenib in VEGFR-2. In the docking of triazole **14** to PD-L1, a hydrogen bond between OH of Tyr56B and NH_2_ is established which would increase the binding force of the ligand to PD-L1 protein.

### 2.2. Synthesis

Triazole derivatives **1**–**26** were achieved by heating at 60 °C for 2–4 h a mixture of the corresponding 1-(azidomethyl)benzene derivative with the corresponding ethynylbenzene derivative in DMF/H_2_O (9:1) in the presence of CuSO_4_·5H_2_O and sodium ascorbate (see Figure 1) [25]. In turn, 1-(azidomethyl)benzene derivatives were achieved by nucleophilic substitution reaction of benzyl bromides or chlorides with sodium azide (see Appendix A). Ethynylbenzene and 4-ethynylaniline were commercially available. The structures of the 26 triazole derivatives, as well as the yields achieved, are indicated in Figure 1.

### 2.3. Biological Evaluation

#### 2.3.1. Study of Cell Viability

The effects of triazole derivatives **1**–**26** on the cell viability were determined by MTT assay, and the IC_50_ values were determined toward the human tumor cell lines HT-29 (colon adenocarcinoma), A-549 (lung adenocarcinoma), and MCF-7 (breast adenocarcinoma), as well as toward the non-tumor cell line HEK-293 (human embryonic kidney cells). In general, compounds were not very active, and we found that phenyl derivatives **1**–**13** exhibited IC_50_ values above 100 μM in all tested cell lines. The IC_50_ values for derivatives **14**–**26** and the reference compounds sorafenib and BMS-8 are shown in Table 1.

Table 1 stands out in that, in general, compounds **14–26** are less active against non-cancer cell line HEK-293. Compounds bearing methoxy or chlorine groups exhibited IC_50_ values above 100 μM in HT-29 and A-549 cell lines, whereas compounds bearing methyl or bromine in their structure exhibited higher antiproliferative activity with IC_50_ values in the low micromolar range. On the other hand, all compounds, except for **18**, exhibited antiproliferative action against MCF-7, with an IC_50_ ranging from 4 to 12 μM.

#### 2.3.2. Effect of Derivatives **14**–**26** on Membrane PD-L1 and VEGFR-2 in Cancer Cell Lines

Based on antiproliferative activity of synthetized derivatives, we decided to study the effect of the amino derivatives **14**–**26** on the expression of both membrane targets PD-L1 and VEGFR-2 (mPD-L1 and mVEGFR-2). The evaluation was performed on three cancer cell lines (HT-29, A-549, and MCF-7) by flow cytometry after 24 h of treatment with the corresponding compounds. Compounds were used at a 100 or 10 μM concentration depending on their IC_50_ value (see Table 1 in Materials and Methods section). The presence of both targets in the membrane was relatively determined by using DMSO-treated cells as a negative control. The action of BMS-8 in membrane PD-L1 was also studied, as well as the influence of sorafenib in membrane VEGFR-2. Table 2 shows the percentage of the detected proteins for each compound referred to as a control (DMSO) on the studied cell lines.

In general, the compounds exhibited a low effect on both membrane targets in all tested cell lines. It is worth highlighting the results obtained for compounds **21**–**23**, bearing a chlorine group, in HT-29 cell line. Thus, *o*-chloro derivative **21** reduced to half the membrane mPD-L1, while *m*-chloro and *p*-chloro derivatives **22** and **23** exhibited a higher effect, yielding detection values around 70% of reduction compared to non-treated cells. In addition, derivatives **22** and **23** were also able to downregulate membrane VEGFR-2 to 65%.

#### 2.3.3. Effect of Derivatives **14**–**26** on Total PD-L1 and c-Myc in Cancer Cell Lines

Based on the results obtained in the previous study, and on the fact that, as small molecules, these compounds may also exhibit effects inside the cell, we decided to evaluate the effect of compounds **14**–**26** on total PD-L1 (that is, both cytosolic and membrane protein (tPD-L1)) and c-Myc, a protein that is located in the nucleus of the cell. The study was carried out again on HT-29, A-549, and MCF-7 cell lines by flow cytometry after 24 h of treatment with the corresponding compounds. The presence of the targets in the whole cell were relatively determined by using DMSO-treated cells as a negative control and BMS-8 as a positive one for PD-L1 (Table 3).

As shown in Table 3, all tested compounds had a similar behavior in both HT-29 and A-549 cell lines, but, clearly it differs from the one exhibited on MCF-7. Thus, regarding PD-L1 in HT-29, compounds bearing a halogen in the structure were the most active ones against total PD-L1, yielding around 40–50% of total target inhibition compared to non-treated cells, while methoxylated analogues **18**–**20** had a moderate effect, outstanding *p*-methoxy derivative **20** which showed a downregulation rate of around 35%. In A-549, *p*-methyl and *p*-methoxy and *o*-bromo derivatives **17**, **20,** and **24**, respectively, were the most active ones, yielding inhibition rates near to 50% compared to non-treated cells. On the other hand, regarding the MCF-7 cell line, only *m*- and *p*-chloro analogues **22** and **23** were active and showed about 40% inhibition of total PD-L1, which is similar to BMS.

Regarding c-Myc, *p*-methoxy derivative **20** is the only one that had any effect in the three cell lines, yielding around 30% of inhibition rates. Besides this one, *p*-chloro analogue **23** in HT-29 and *p*-bromo derivative **26** in A-549 had a mild effect on c-Myc, with 20% of target inhibition as compared to non-treated cells. It is interesting to highlight that all compounds, except for the *o*-methyl derivative **15**, were able to downregulate c-Myc to 50–60% in MCF-7.

#### 2.3.4. Study of Cellular PD-1/PD-L1 Blocking Activity in Co-Cultures: Effect on Cancer Cell Viability

Cellular PD-1/PD-L1 blocking activity was studied in a co-culture system with PD-1 Jurkat T cells stimulated by interferon γ and PD-L1-expressing cancer cells. Thereby, we selected chloro derivatives **21**–**23** to study the proliferation of cancer cells co-cultured in the presence of PD-1 expressing Jurkat T cells in order to evaluate whether the observed PD-L1 inhibition is translated to the blockage of PD-1/PD-L1 system. Compounds which showed the best PD-L1 inhibition rates in previous studies were evaluated. Thus, tumor cells HT-29 were treated for 24 h with the selected compounds at 100 μM in the presence of Jurkat T cells and then living cells were counted by using cytometry and a Neubauer chamber. Table 4 shows the relative number of living cells from both kinds of populations, HT-29 and Jurkat T cells, related to non-treated samples.

From data provided in Table 4, it can be concluded that the tested compounds are able to inhibit HT-29 cell proliferation in co-culture at rates similar to the effect exerted by BMS-8. Interestingly, in the HT-29 cell line this effect correlates to those obtained in the previous study of protein inhibition in HT-29 cells (see entries 8, 9, and 10 in Table 3).

#### 2.3.5. Study of the Direct Interaction with PD-L1 Protein

The affinity of the chloro derivatives **21**, **22**, and **23** for PD-L1, which were the most active ones targeting this protein, was studied by thermal shift assay [13]. This assay measures the melting point (Tm) of the target protein which is a proxy of its stability that can be altered by the binding of a ligand. A change in the melting point relative to the unliganded form implies that a ligand has bound itself to the targeted protein. According to the fundamentals of this technique, an increase in Tm means that the binding of the ligand leads to a stabilization of the target protein, while a decrease in Tm is probably due to the fact that the binding between the ligand and its target leads to a destabilization of the protein. The results achieved are indicated in Table 5. In this case, the tested compounds were able to change the PD-L1 melting temperature. The three compounds provoke a shift in the protein melting point that correlates with results obtained in the previous study of protein inhibition in co-cultures of HT-29 and Jurkat T cells (see Table 4) in which compounds **21**–**23** were the most active at inhibiting this target.

#### 2.3.6. Study of PD-1/PD-L1 Blocking Activity by Competitive ELISA Assay

After assessing the affinity of the chloro derivatives **21**, **22**, and **23** for PD-L1 by thermal shift assay, we explored their effect on PD-1/PD-L1 binding by competitive ELISA [26].

This assay is based on the extremely affinity of PD-1 and PD-L1, and we tested it by using a concentration of 150 μM for the three derivatives.

The results are depicted in Figure 4, and we observed that the tested compounds were able to inhibit 30–40% of the PD-1/PD-L1 interaction.

#### 2.3.7. Study of Potential Kinase Inhibitory Activity by ADP Assay

Finally, to assess the VEGFR-2 inhibitory activity of the selected compounds, namely **21**, **22**, and **23**, we performed an ADP assay. This is a very useful technique that measures kinase activity by monitoring ADP accumulation in cell cultures instead of ATP concentrations [27].

For the assay, we used HT-29 cells that were previously treated with a 100 μM concentration of **21**, **22**, and **23**, and we used DMSO as a negative control. The results presented in Table 6 show that these compounds managed to inhibit VEGFR-2 kinase activity.

#### 2.3.8. Study of Antivascular and Antiangiogenic Effect on HMEC-1 by Microtubes Formation on Matrigel^®^ Assay

The antivascular and antiangiogenic activities of the selected compounds, namely **21**, **22**, and **23**, were settled by 3D Matrigel cell cultures. First, we studied the effect on a microvessel network from HMEC-1 cells. HMEC-1 cells were on Matrigel, and after 24 h, when organized capillary networks were formed, 100 μM of the corresponding compounds, **21**–**23,** was added. After another 24 h, we compared the treated networks with the one that received no treatment (see Figure 5a–c). The treatment with these compounds resulted in a disturbance of the formed microvasculature-like network, with disruption in the intercellular connections.

Then we went on to study the effect of these selected compounds on the formation of new capillary networks. Again, HMEC-1 cells were seeded on Matrigel and treated with compounds **21**, **22**, and **23** at 100 μM. After 24 h of the corresponding treatments, we compared the cultures with non-treated cells, and the results are depicted in Figure 5d–f. As it can be observed, all compounds interfered with the formation of new microvessel networks, with the majority of the cells forming tiny clumps, and with most cells forming tiny clusters rather than establishing any organized intercellular connections.

## 3. Discussion

The simpler phenyl triazoles **1**–**13** showed moderate action on cell viability in the high micromolar range in all tested cell lines. On the other hand, aniline derivatives were slightly more active than phenyl ones against most tested cancer cell lines. It is worth mentioning that all of these compounds exhibit higher action against MCF-7 than on the rest of tested cancer cell lines. In this case, compounds **14**–**26** exhibited IC_50_ values ranging from 4 to 12 μM, which are even lower than the reference compounds sorafenib and BMS-8. In the rest of the cancer cell lines, HT-29 and A-549, compounds bearing methyl and bromine groups exhibited IC_50_ values around 10 μM, whereas those with methoxy and chlorine groups showed IC_50_ values above 100 μM. It is interesting to note that all derivatives showed good selectivity toward cancer cells, except for compound **17**, exhibiting higher IC_50_ values for the HEK-293 cell line.

From these results, we could assume that electron donating groups such as amino group in aromatic ring 1 (see Figure 6) enhance the antiproliferative action of these small molecules, whereas groups with a negative inductive effect—that is, electron-withdrawing groups, such a methoxy—in ring 2 decrease antiproliferative activity. As the electron-releasing tendency of functional groups in R_2_ increases, cells became more susceptible to damage.

Considering that, in general, all compounds were clearly more active against MCF-7 cell viability than on the rest of the tested cell lines, we assume that the mechanism for their action is different depending on cell lines. This assumption correlated with the results obtained in the studies on their biological effect on our targets: PD-L1, VEGFR-2, and c-Myc.

In this sense, we observed that the effect on c-Myc protein was practically negligible in HT-29 and A-549; however, the rate of inhibition in MCF-7 was around 45% for most of the tested derivatives (see Table 3). We assume that these results could explain the higher activity observed for all derivatives on MCF-7 cell viability inhibition compared to the rest of studied cell lines (see Table 1). These singular results obtained for MCF-7 could also be attributed to the differential sensitivity profile of every cell line due to the differences in their biochemical and metabolic characteristics. For example, MCF-7 is the only of the tested cell lines that produces insulin-like growth factor binding protein (IGFBP), a protein that promotes tumorigenesis [28,29]. In the future, we will try to elucidate which target is responsible for the differences in the response of MCF-7 when treated with our derivatives, but currently it goes beyond the aim of the present work.

Focusing on PD-L1 and VEGFR-2, we have found that the HT-29 cell line is the most sensitive to the treatment with the tested compounds. In general, compounds exert a higher effect on PD-L1 than in VEGFR-2 in all tested cell lines, and those compounds bearing halogens in their structure are more active than those lacking halogens. Moreover, the chloro derivatives exhibited higher action against both targets than bromo ones.

As regards membrane PD-L1, all three chloro derivatives inhibit more than half of the membrane target. Specially, *m*-chloro derivative **22** was able to inhibit around 75% of membrane PD-L1. In addition, *m*-chloro and *p*-chloro derivatives **22** and **23** also inhibited around a 40% of membrane VEGFR-2 with respect to untreated cells. Moreover, when we studied the effect of the synthetic compounds on total PD-L1, we found that not only chloro but also bromo derivatives exhibited good activity, showing around 60% of relative expression in total PD-L1. The fact that bromo compounds **24**–**26** had lower action on membrane than on total PD-L1, while chloro derivatives were more active on membrane than on total PD-L1, can be attributed to a higher lipophilicity of bromine as regards chlorine derivatives, thus facilitating the entrance of the bromo derivatives into the cell and therefore the action on the cytosolic and nuclear PD-L1 would not be so depending on this substituent.

Our hypothesis that the observed inhibition of PD-L1 was contributing to the blockage of the PD-1/PD-L1 axis was corroborated by performing two independent assays. First we studied the effect of the most active compounds against PD-L1 (the chloro derivatives **21**, **22**, and **23**) on HT-29 cell viability in the presence of PD-1-expressing Jurkat T-cells. As expected, these compounds reduced tumor cell viability by half compared to the negative control (untreated cells) and did not affect the viability of the Jurkat T cells (see Table 4 and Figure 7). Then the PD-1/PD-L1 blocking activity of these compounds was corroborated by competitive ELISA (see Figure 3 and Figure 7).

All the results are in correlation with our hypothesis that these derivatives were contributing to the blockage of the PD-1/PD-L1 axis. Figure 7 shows the results obtained with chloro derivatives **21**, **22**, and **23**.

To ascertain the higher action of chloro derivatives on PD-L1, we also performed a protein thermal shift assay and completed docking studies. Thus, as shown in Table 5, chloro derivatives exhibited a displacement in a melting temperature of PD-L1 higher than 4 °C, similar to our positive control BMS-8. This is in accordance with a good affinity of these derivatives toward the biological target, as shown in Figure 8a. This figure depicts the docking superposition of compound **23** (orange) with BMS-202 (gray) in PD-L1 protein. It can be appreciated that the aromatic ring 1 of triazole **23** is interacting with Tyr56 from chain B of PD-L1 through π-stacking, while Cl atom in ring 2 is showing a T stacking interaction with Tyr56 of chain A. For the sake of comparison, the docking superposition of BMS-202 (gray) with compound **20** (blue), bearing a methoxy group in *para* position in ring 2, is shown in Figure 5b. In this case, the larger size of the methoxy group shifts molecule **20** to the left, thus losing the π-stacking interaction with Tyr56 of chain B, which could explain the lesser activity of this molecule.

As regards VEGFR-2, chloro compounds **21**–**23** exhibited a moderate action on the presence of membrane VEGFR-2, but when we checked the kinase activity of treated cells, we corroborated that they inhibit this action more than 60% compared to non-treated cells. Moreover, the selected triazoles, **21**–**23**, were active in inhibiting the formation of new capillary tubes from HMEC-1, and they were also able to destroy the microvessel networks. Therefore, compounds **21**–**23** are promising antivascular and antiangiogenic agents with immunomodulation properties.

In conclusion, the work we present herein opens the door for future studies with structures based on this anilinyl triazole scaffold for the design of multitarget agents exhibiting antivascular, antiangiogenic, and anti-PD-L1 activities. Moreover, the presence of the amino group can be used for their binding to nanoparticles or other molecules in order to enhance their ADME properties.

## 4. Materials and Methods

### 4.1. Chemistry

#### 4.1.1. General Procedures

^1^H and ^13^C NMR spectra were measured at 25 °C. The signals of the deuterated solvent (DMSO_D6_) were taken as the reference. Multiplicity assignments of ^13^C signals were made by means of the DEPT pulse sequence. Complete signal assignments in ^1^H and ^13^C NMR spectra were made with the aid of 2D homo- and heteronuclear pulse sequences (COSY, HSQC, and HMBC). High-resolution mass spectra were recorded by using electrospray ionization–mass spectrometry (ESI–MS). Experiments which required an inert atmosphere were carried out under dry N_2_ in oven-dried glassware. Commercially available reagents were used as received.

#### 4.1.2. Experimental Procedure for the Synthesis of Triazole Compounds **1**–**26**

A solution of the corresponding 1-(azidomethyl)benzene derivative (3 mmol) with the corresponding ethynylbenzene derivative (2 mmol) in DMF/H_2_O (9:1, 50 mL) was heated at 60 °C for 2–4 h in the presence of CuSO_4_·5H_2_O (0.25 mmol) and sodium ascorbate (0.25 mmol). Then the reaction mixture was poured onto brine, and the aqueous phase was extracted three times with Ethyl Acetate. The collected organic phases were washed with brine and dried over anhydrous Na_2_SO_4_. Filtration and removal of the solvent under vacuum afforded a residue that was purified on column chromatography, using silica gel as stationary phase and a mixture of Hexanes:Ethyl Acetate (1:1; 4:6; 3:7) as mobile phase.

1-benzyl-4-phenyl-1*H*-1,2,3-triazole **(1):** yield, 52%, white solid, m.p. 127–129 °C; ^1^H NMR (400 MHz, CDCl_3_): δ 7.71 (dd, *J* = 7.3, 1.3 Hz, 2H), 7.58 (s, 1H), 7.33–7.16 (m, 8H), 5.44 (s, 2H); ^13^C NMR (100 MHz, CDCl_3_): 148.1 (C), 134.6 (C), 130.5 (C), 129.0 (CH), 128.7 (CH), 128.6 (CH), 128.0 (CH), 127.9 (CH), 125.6 (CH), 119.5 (CH), 54.1 (CH_2_); HR ESMS *m*/*z* 236.1183 (M + H^+^). Calc. C_15_H_13_N_3_: 235.11.

1-(2-methylbenzyl)-4-phenyl-1*H*-1,2,3-triazole **(2)**: yield, 65%; ^1^H NMR (400 MHz, CDCl_3_): δ 7.76 (dd, *J* = 7.0, 1,2 Hz, 2H), 7.51 (s, 1H), 7.37 (td, *J* = 7,0, 1,2 Hz, 2H), 7.31–7.16 (m, 5H), 5.56 (s, 2H), 2.29 (s, 3H); ^13^C NMR (100 MHz, CDCl_3_): 148.0 (C), 137.0 (C), 132.5 (C), 131.1 (CH), 130.6 (CH), 129.4 (C), 129.2 (CH), 128.8 (CH), 128.1 (CH), 126.7 (CH), 125.7 (CH) 119.2 (CH), 52.6 (CH_2_), 19.0 (CH_3_); HR ESMS *m*/*z* 250.1339 (M + H^+^). Calc. C_16_H_15_N_3_: 249.13.

1-(3-methylbenzyl)-4-phenyl-1*H*-1,2,3-triazole **(3)**: yield, 43%; ^1^H NMR (400 MHz, CDCl_3_): δ 7.71 (dd, *J* = 7.0, 1.3 Hz, 2H), 7.59 (s, 1 H), 7.30–7.25 (m, 2H), 7.22–6.98 (m, 7H), 5.38 (s, 2H), 2.22 (s, 3H); ^13^C NMR (100 MHz, CDCl_3_): 147.9 (C), 138.8 (C), 134.5 (C), 130.5(C), 129.3 (CH), 128.8 (CH), 128.6 (CH), 128.6 (CH), 127.9 (CH), 125.5 (CH), 124.9 (CH), 119.5 (CH), 54.0 (CH_2_), 21.1 (CH_3_); HR ESMS *m*/*z* 250.1347 (M + H^+^). Calc. C_16_H_15_N_3_: 249.13.

1-(4-methylbenzyl)-4-phenyl-1*H*-1,2,3-triazole **(4)**: yield, 55%; ^1^H NMR (400 MHz, CDCl_3_): δ 7.70 (dd, *J* = 7.0, 1.2 Hz, 2H), 7.54 (s, 1H), 7.32–7.07 (m, 7H), 5.42 (s,2H), 2.26 (s, 3H); ^13^C NMR (100 MHz, CDCl_3_): 148.1 (C), 138.6 (C), 131.6 (C), 130.6 (C), 129.7 (CH), 128.7 (CH), 128.1 (CH), 128.0 (CH), 125.6 (CH), 119.4 (CH), 54.0 (CH_2_), 21.1 (CH_3_); HR ESMS *m*/*z* 250.1342 (M + H^+^). Calc. C_16_H_15_N_3_: 249.13.

1-(2-methoxybenzyl)-4-phenyl-1*H*-1,2,3-triazole **(5)**: yield, 52%; ^1^H NMR (400 MHz, CDCl_3_): δ 7.78 (dd, *J* = 7.0, 1.2 Hz, 2H), 7.68 (s, 1H), 7.40–7.18 (m, 5H), 6.96–6.90 (m, 2H), 5.57 (s, 2H), 3.86 (s, 3H); ^13^C (100 MHz, CDCl_3_): 157.2 (C), 147.7 (C), 130.8 (C), 130.3 (CH), 128.7 (CH), 127.9 (CH), 125.7 (CH), 123.0 (C), 121.0 (CH), 119.7 (CH), 110.8 (CH), 55.5 (CH_2_), 49.2 (CH_3_); HR ESMS *m*/*z* 266.1296 (M + H^+^). Calc. C_16_H_15_N_3_O: 265.12.

1-(3-methoxybenzyl)-4-phenyl-1*H*-1,2,3-triazole **(6)**: yield, 63%; ^1^H NMR (400 MHz, CDCl_3_): δ 7.80 (dd, *J* = 7.0, 1.2 Hz, 2H), 7.67 (s, 1H), 7.43–7.38 (m, 2H), 7.33–7.25 (m, 3H), 6.92–6.83 (m, 2H), 5.54 (s, 2H), 3.79 (s, 3H); ^13^C (100 MHz, CDCl_3_): 160.2 (C), 148.2 (C), 136.1 (C), 130.5 (C), 130.2 (CH), 128.8 (CH), 128.1 (CH), 125.7 (CH), 120.2 (CH), 119.5 (CH), 114.2 (CH), 113.6 (CH), 55.3 (CH_2_), 54.2 (CH_3_); HR ESMS *m*/*z* 266.1297 (M + H^+^). Calc. C_16_H_15_N_3_O: 265.12.

1-(4-methoxybenzyl)-4-phenyl-1*H*-1,2,3-triazole **(7)**: yield, 50%; ^1^H NMR (400 MHz, CDCl_3_): δ 7.80 (dd, *J* = 7.1, 1.3 Hz, 2H), 7.64 (s, 1H), 7.4 (dt, *J* = 7.1, 1,2 Hz, 2H), 7.34–7.24 (m, 3H), 6.93–6.90 (m, 1H), 6.84–6.83 (t, 1H), 5.50 (s, 2H), 3.81 (s,3H); ^13^C NMR (100 MHz, CDCl_3_): 159.9 (C), 148.1 (C), 130.6 (C), 129.6 (CH), 128.7 (CH), 128.1 (CH), 126.6 (C), 125.7 (CH), 119.3 (CH), 114.5 (CH), 55.3 (CH_2_), 53.7 (CH_3_); HR ESMS *m*/*z* 266.1294 (M + H^+^). Calc. C_16_H_15_N_3_O: 265.12.

1-(2-chlorobenzyl)-4-phenyl-1*H*-1,2,3-triazole **(8)**: yield, 67%, white solid, m.p. 86–88 °C; ^1^H NMR (400 MHz, CDCl_3_): δ 7.75 (dd, *J* = 8.2, 1.4 Hz, 2H), 7.69 (s, 1H), 7.38 (dd, *J* = 8.0, 1.2 Hz, 1H), 7.34 (t, *J* = 7.4 Hz, 2H), 7.15–7.25 (m, 4H), 5.65 (s, 2H); ^13^C NMR (100 MHz, CDCl_3_): δ 148.2 (C), 133.5 (C), 132.6 (C), 130.5 (C), 130.3 (CH), 130.3 (CH), 130.0 (CH), 128.8 (CH), 128.2 (CH), 127.7 (CH), 125.8 (CH), 119.8 (CH), 51.5 (CH_2_); HR ESMS *m*/*z* 270.0798 (M + H^+^). Calc. C_15_H_12_ClN_3_: 269.07.

1-(3-chlorobenzyl)-4-phenyl-1*H*-1,2,3-*triazole*
**(9)**: yield, 59%, white solid, m.p. 108–110 °C; ^1^H NMR (400 MHz, CDCl_3_): δ 7.74 (d, *J* = 8.4, 1.2 Hz, 2H), 7.62 (s, 1H), 7.34 (t, *J* = 7.4 Hz, 2H), 7.24–7.29 (m, 4H), 7.11 (dt, *J* = 6.8, 1.8 Hz, 1H), 5.48 (s, 2H); ^13^C NMR (100 MHz, CDCl_3_): δ 148.6 (C), 136.7 (C), 135.1 (C), 130.6 (CH), 130.5 (C), 129.1 (CH), 128.9 (CH), 128.4 (CH), 128.2 (CH), 126.1 (CH), 125.8 (CH), 119.6 (CH), 53.6 (CH_2_); HR ESMS *m*/*z* 270.0798 (M + H^+^). Calc. C_15_H_12_ClN_3_: 269.07.

1-(4-chlorobenzyl)-4-phenyl-1*H*-1,2,3-triazole **(10)**: yield, 65%, white solid, m.p. 141–144 °C; ^1^H NMR (400 MHz, CDCl_3_): δ 7.70 (d, *J* = 7.2 Hz, 2H), 7.59 (s, 1H), 7.30 (t, *J* = 7.4 Hz, 2H), 7.20–7.26 (m, 3H), 7.14 (d, *J* = 8.4 Hz, 2H), 5.43 (s, 2H); ^13^C NMR (100 MHz, CDCl_3_): δ 148.4 (C), 134.8 (C), 133.3 (C), 130.4 (C), 129.4 (CH), 129.4 (CH), 128.9 (CH), 128.3 (CH), 125.7 (CH), 119.5 (CH), 53.5 (CH_2_); HR ESMS *m*/*z* 270.0798 (M + H^+^). Calc. C_15_H_12_ClN_3_: 269.07.

1-(2-bromobenzyl)-4-phenyl-1*H*-1,2,3-triazole **(11)**: yield, 47%,white solid, m.p. 100–103 °C; ^1^H NMR (400 MHz, CDCl_3_): δ 7.75 (d, *J* = 8.0 Hz, 2H), 7.71 (s, 1H), 7.57 (d, *J* = 8.0 Hz, 1H), 7.34 (t, *J* = 7.8 Hz, 2H), 7.25 (t, *J* = 7.8 Hz, 2H), 7.14 (m, 2H), 5.65 (s, 2H); ^13^C NMR (100 MHz, CDCl_3_): δ 148.2 (C), 134.3 (C), 133.3 (CH), 130.4 (C), 130.4 (CH), 130.3 (CH), 128.8 (CH), 128.3 (CH), 128.2 (CH), 125.8 (CH), 123.4 (C), 119.8 (CH), 53.9 (CH_2_); HR ESMS *m*/*z* 314.0293 (M + H^+^). Calc. C_15_H_12_BrN_3_: 313.02.

1-(3-bromobenzyl)-4-phenyl-1*H*-1,2,3-triazole **(12)**: yield, 64%, white solid, m.p. 96–99 °C; ^1^H NMR (400 MHz, CDCl_3_): δ 7.73 (d, *J* = 8.4, 2H), 7.62 (s, 1H), 7.42 (t, *J* = 7.8, 1H), 7.39 (s, 1H), 7.33 (t, *J* = 7.4, 2H), 7.25 (t, *J* = 7.4, 1H), 7.16 (m, 2H), 5.46 (s, 2H); ^13^C NMR (100 MHz, CDCl_3_): δ 148.5 (C), 137.0 (C), 132.1 (CH), 131.1 (CH), 130.8 (CH), 130.4 (C), 128.9 (CH), 128.4 (CH), 126.6 (CH), 125.8 (CH), 123.2 (C), 119.6 (CH), 53.5 (CH_2_); HR ESMS *m*/*z* 314.0293 (M + H^+^). Calc. C_15_H_12_BrN_3_: 313.02.

1-(4-bromobenzyl)-4-phenyl-1*H*-1,2,3-triazole **(13)**: yield, 65%, white solid, m.p. 153–155 °C; ^1^H NMR (400 MHz, CDCl_3_): δ 7.73 (dd, *J* = 8.3, 1.2 Hz, 2H), 7.59 (s, 1H), 7.45 (d, *J* = 8.4 Hz, 2H), 7.34 (t, *J* = 7.4 Hz, 2H), 7.25 (tt, *J* = 7.4 Hz, 1.2, 1H), 7.12 (d, *J* = 8.4, 2H), 5.47 (s, 2H); ^13^C NMR (100 MHz, CDCl_3_): δ 148.4 (C), 133.7 (C), 132.4 (CH), 130.4 (C), 129.7 (CH), 128.8 (CH), 128.3 (CH), 125.7 (CH), 123.0 (C), 119.4 (CH), 53.6 (CH_2_); HR ESMS *m*/*z* 314.0293 (M + H^+^). Calc. C_15_H_12_BrN_3_: 313.02.

4-(1-benzyl-1*H*-1,2,3-triazol-4-yl)aniline **(14)**: yield, 34%, brownish solid, m.p. 180–181 °C; ^1^H NMR (300 MHz, DMSO-d6): δ 8.41 (s, 1H), 7.48 (d, *J* = 8.7 Hz, 2H), 7.42–7.28 (m, 5H), 5.57 (s, 2H); ^13^C NMR (75 MHz, DMSO-d6): δ 148.9 (C), 147.6 (C), 136.2 (C), 128.7 (CH), 128.0 (CH), 127.8 (CH), 126.1 (CH), 119.3 (CH), 118.4 (C), 114.0 (CH), 52.8 (CH_2_); HR ESMS *m*/*z* 251.1297 (M + H^+^). Calc. C_15_H_14_N_4_: 250.12.

4-(1-(2-methylbenzyl)-1*H*-1,2,3-triazol-4-yl)aniline **(15)**: yield, 25%, brownish solid, m.p. 131–133 °C; ^1^H NMR (400 MHz, CDCl_3_): δ 7.78 (d, 8.8 Hz, 2H), 7.32 (s, 1H), 7.23–7.05 (m, 4H), 6.59 (d, *J* = 8.8 Hz, 2H), 5.44 (s, 2H), 2.21 (s, 3H); ^13^C NMR (100 MHz, CDCl_3_): δ 148.4 (C), 146.6 (C), 137 (C), 132.8 (CH), 131.1 (CH), 129.4 (CH), 129.1 (CH), 126.9 (CH), 126.7 (CH), 121.1 (C), 118.1 (CH), 115.3 (CH), 52.4 (CH_2_), 19.1 (CH_3_); HR ESMS *m*/*z* 265,1453 (M + H^+^). Calc. C_16_H_16_N_4_: 264.14.

4-(1-(3-methylbenzyl)-1*H*-1,2,3-triazol-4-yl)aniline **(16)**: yield, 42%, brownish solid, m.p. 128–129 °C; ^1^H NMR (300 MHz, DMSO-d6): δ 8.30 (s, 1H), 7.48 (d, *J* = 8.8 Hz, 2H), 7.27 (t, *J* = 7.8 Hz, 1H), 7.20–7.08 (m, 3 H), 6.60 (d, *J* = 8.8 Hz, 2H), 5.54 (s, 2H), 5.22 (s, 2H), 2.23 (s, 3H); ^13^C NMR (75 MHz, DMSO): δ 149.1 (C), 147.6 (C), 137.9 (C), 136.1 (C), 128.7 (CH), 128.6 (CH), 128.4 (CH), 126.1 (CH), 124.9 (CH), 119.2 (CH), 118.3 (C), 113.9 (CH), 52.8 (CH_2_), 20.9 (CH_3_); HR ESMS *m*/*z* 265,1453 (M + H^+^). Calc. C_16_H_16_N_4_: 264.14.

4-(1-(4-methylbenzyl)-1*H*-1,2,3-triazol-4-yl)aniline **(17)**: yield, 58%, brownish solid, m.p. 130–132 °C; ^1^H NMR (300 MHz, DMSO-d6): δ 8.23 (s, 1H), 7.48 (d, *J* = 8.7 Hz, 2H), 7.23 (d, *J* = 8.4 Hz, 2H), 7.19 (d, *J* = 8.4 Hz, 2H), 6.59 (d, *J* = 8.7 Hz, 2H), 5.52 (s, 2H), 5.21 (s, 2H), 2,29 (s, 3H); ^13^C NMR (75 MHz, DMSO-d6): δ 148.6 (C), 147.6 (C), 137.3 (C), 133.1 (C), 129.2 (CH), 127.8 (CH), 126.1 (CH), 119.1 (CH), 118.3 (C), 113.9 (CH), 52.6 (CH_2_), 20.5 (CH_3_); HR ESMS *m*/*z* 265,1453 (M + H^+^). Calc. C_16_H_16_N_4_: 264.14.

4-(1-(2-methoxybenzyl)-1*H*-1,2,3-triazol-4-yl)aniline **(18)**: yield, 52%, brown solid, m.p. 166–167 °C; ^1^H NMR (400 MHz, MeOD): δ 7.94 (s, 1H), 7.49 (d, *J* = 8 Hz, 2H), 7.32 (td, *J* = 8, 4 Hz, 1H), 7.19 (dd, *J* = 8, 4 Hz, 1H), 7.00(d, *J* = 12 Hz, 1H), 6.93 (td, *J* = 8, 1 Hz, 2H), 6.73 (d, *J* = 8 Hz, 2H), 5.55 (s, 2H), 3.85 (s, 3H); ^13^C NMR (100 MHz, MeOD): δ 158.6 (C), 149.3 (C), 149.1 (C), 131.2 (CH), 130.7 (CH), 127.5 (CH), 124.3 (C), 121.7 (CH), 120.8 (C), 120.5 (CH), 116.2 (CH), 111.8 (CH), 55.8 (CH_3_), 50.2 (CH_2_); HR ESMS *m*/*z* 281,1404 (M + H^+^). Calc. C_16_H_16_N_4_O: 280.13.

4-(1-(3-methoxybenzyl)-1*H*-1,2,3-triazol-4-yl)aniline **(19)**: yield, 20%, brown solid, m.p. 103–105 °C; ^1^H NMR (400 MHz, MeOD): δ 8.05 (s, 1H), 7.51 (d, *J* = 8 Hz, 2H), 7.30–7.22 (m, 1H), 6.92–6.85 (m, 3H), 6.73 (d, *J* = 8 Hz, 2H), 5.53 (s, 2H), 3.75 (s, 3H); ^13^C NMR (100 MHz, MeOD): δ 161.8 (C), 150.2 (C), 149.7 (C), 138.5 (C), 131.4 (CH), 128.0 (CH), 121.4 (CH), 121.3 (C), 120.9 (CH), 116.6 (CH), 115.3 (CH), 114.8 (CH), 56.0 (CH_2_), 55.0 (CH_3_); HR ESMS *m*/*z* 281,1397 (M + H^+^). Calc. C_16_H_16_N_4_O: 280.13.

4-(1-(4-methoxybenzyl)-1*H*-1,2,3-triazol-4-yl)aniline **(20)**: yield, 38%, brown solid, m.p. 132–134 °C; ^1^H NMR (400 MHz, MeOD): δ 8.01 (s, 1H), 7.50 (d, *J* = 8 Hz, 2H), 7.30 (d, *J* = 8 Hz, 2H), 6.92 (d, *J* = 8 Hz, 2H), 6.74 (d, *J* = 8 Hz, 2H), 5.50 (s, 2H), 3.77 (s, 3H); ^13^C NMR (100 MHz, MeOD): δ 161.4 (C), 149.9 (C), 149.4 (C), 130.6 (CH), 128.8 (C), 127.7 (CH), 121.1 (C), 120.3 (CH), 116.4 (CH), 115.4 (CH), 55.7 (CH_3_), 54.5 (CH_2_); HR ESMS *m*/*z* 281,1404 (M + H^+^). Calc. C_16_H_16_N_4_O: 280.13.

4-(1-(2-chlorobenzyl)-1*H*-1,2,3-triazol-4-yl)aniline **(21)**: yield, 39%, brown solid, m.p. 112–115 °C; ^1^H NMR (400 MHz, MeOD): δ 8.03 (s, 1H), 7.52 (d, *J* = 8 Hz, 2H), 7.44 (dd, *J* = 8, 4 Hz, 1H), 7.36–7.26 (m, 2H), 7.22 (d broad, *J* = 8 Hz, 1H), 6.74 (d, *J* = 8 Hz, 2H), 5.68 (s, 2H); ^13^C NMR (100 MHz, MeOD): δ 150.0 (C), 149.6 (C), 134.8 (C), 134.5 (C), 131.6 (CH), 131.5 (CH), 131.1 (CH), 128.8 (CH), 128.0 (CH), 121.2 (CH), 121.1 (C), 116.6 (CH), 52.5 (CH_2_); HR ESMS *m*/*z* 285,0912 (M + H^+^). Calc. C_15_H_13_ClN_4_: 284.08.

4-(1-(3-chlorobenzyl)-1*H*-1,2,3-triazol-4-yl)aniline **(22)**: yield, 40%, brownish solid, m.p. 143–144 °C; ^1^H NMR (400 MHz, MeOD): δ 8.12 (s, 1H), 7.53 (d, *J* = 8 Hz, 2H), 7.40–7.33 (m, 3H), 7.29–7.24 (m, 1H), 6.75 (d, *J* = 8 Hz, 2H), 5.58 (s, 2H); ^13^C NMR (100 MHz, MeOD): δ 150.1 (C), 149.6 (C), 139.2 (C), 135.7 (C), 131.5 (CH), 129.6 (CH), 129.0 (CH), 127.7 (CH), 127.4 (CH), 120.9 (C), 120.6 (CH), 116.4 (CH), 54.0 (CH_2_); HR ESMS *m*/*z* 285,0908 (M + H^+^). Calc. C_15_H_13_ClN_4_: 284.08.

4-(1-(4-chlorobenzyl)-1*H*-1,2,3-triazol-4-yl)aniline **(23)**: yield, 71%, brownish solid, m.p. 164–166 °C; ^1^H NMR (400 MHz, MeOD): δ 8.09 (s, 1H), 7.52 (d, *J* = 8 Hz, 2H), 7.39 (d, *J* = 8 Hz, 2H), 7.33 (d, *J* = 8 Hz, 2H), 6.75 (d, *J* = 8 Hz, 2H), 5.58 (s, 2H); ^13^C NMR (100 MHz, MeOD): δ 150.3 (C), 149.8 (C), 135.9 (C), 135.6 (C), 130.9 (CH), 130.3 (CH), 127.9 (CH), 121.2 (C), 120.8 (CH), 116.5 (CH), 54.3 (CH_2_); HR ESMS *m*/*z* 285,0903 (M + H^+^). Calc. C_15_H_13_ClN_4_: 284.08.

4-(1-(2-bromobenzyl)-1*H*-1,2,3-triazol-4-yl)aniline **(24)**: yield, 52%, brown solid, m.p. 133–134 °C; ^1^H NMR (300 MHz, MeOD): δ 8.04 (s, 1 H), 7.63 (dd, *J* = 7.9, 1.2 Hz, 1 H), 7.51 (d, *J* = 8.7 Hz, 2 H), 7.34 (td, *J* = 7.5, 1.2 Hz, 1H), 7.24 (td, *J* = 7.5, 1.7 Hz, 1H), 7.17 (dd, *J* = 7.5, 1.7 Hz, 1H), 6.73 (d, *J* = 8.7, 2H), 5.68 (s, 2H); ^13^C NMR (75 MHz, MeOD): δ 149.8 (C), 149.6 (C), 136.0 (C), 134.3 (CH), 131.5 (CH), 131.4 (CH), 129.3 (CH), 127.8 (CH), 124.4 (C), 121.1 (CH), 120.9 (C), 116.4 (CH), 54.9 (CH_2_); HR ESMS *m*/*z* 329,0402 (M + H^+^). Calc. C_15_H_13_BrN_4_: 328.03.

4-(1-(3-bromobenzyl)-1*H*-1,2,3-triazol-4-yl)aniline **(25)**: yield, 73%, brownish solid, m.p. 134–135 °C; ^1^H NMR (300 MHz, DMSO-d6): δ 8.35 (s, 1 H), 7.60–7.52 (m, 2H 1H), 7.40–7.29 (m, 2H), 7.49 (d, *J* = 9 Hz, 2 H), 6.60 (d, *J* = 9 Hz, 2H), 5.61 (s, 2H), 5.24 (s, 2H); ^13^C NMR (75 MHz, DMSO): δ 148.6 (C), 147.7 (C), 138.8 (CH), 131.0 (2 x CH), 130.6 (CH), 126.9 (CH), 126.2 (CH), 121.8 (C), 119.4 (C), 118.1 (C), 113.9 (CH), 52.0 (CH_2_); HR ESMS *m*/*z* 329,0402 (M + H^+^). Calc. C_15_H_13_BrN_4_: 328.03.

4-(1-(4-bromobenzyl)-1*H*-1,2,3-triazol-4-yl)aniline **(26)**: yield, 88%, brownish solid, m.p. 169–170 °C; ^1^H NMR (300 MHz, C_3_D_6_O): δ 8.08 (s, 1H), 7.53 (d, *J* = 8.7 Hz, 2H), 7.52 (d, *J* = 8.7 Hz, 2H), 7.29 (d, *J* = 9 Hz, 2H), 6.66 (d, *J* = 9 Hz, 2H), 5.58 (s 2H), 4.71 (s, 2H); ^13^C NMR (75 MHz, C_3_D_6_O): δ 148.3 (C), 148.2 (C), 135.7 (C), 131.7 (CH), 129.9 (CH), 126.3 (CH), 121.5 (C), 119.7 (C), 118.5 (CH), 114.2 (CH), 52.4 (CH_2_); HR ESMS *m*/*z* 329,0402 (M + H^+^). Calc. C_15_H_13_BrN_4_: 328.03.

### 4.2. Biological Studies

#### 4.2.1. Cell Culture

Cell culture media were purchased from Gibco (Grand Island, NY, USA). Fetal bovine serum (FBS) was obtained from Harlan-Seralab (Belton, UK). Supplements and other chemicals not listed in this section were obtained from Sigma Chemical Co. (St. Louis, MO, USA). Plastics for cell culture were supplied by Thermo Scientific BioLite. All tested compounds were dissolved in DMSO at a concentration of 20 mM and stored at −20 °C until use.

HT-29, A549, MCF-7, and HEK-293 cell lines were maintained in Dulbecco’s modified Eagle’s medium (DMEM) containing glucose (1 g/L), glutamine (2 mM), penicillin (50 μg/mL), streptomycin (50 μg/mL), and amphotericin B (1.25 μg/mL), supplemented with 10% FBS. 

#### 4.2.2. Cell Proliferation Assay

In 96-well plates, 3 × 10^3^ (A549 and HEK-293) or 5 × 10^3^ (HT-29, MCF-7) cells per well were incubated with serial dilutions of the tested compounds in a total volume of 100 μL of their growth media. The 3-(4,5-dimethylthiazol-2-yl)-2,5-diphenyltetrazolium bromide (MTT; Sigma Chemical Co.) dye reduction assay in 96-well microplates was used. After 2 days of incubation (37 °C, 5% CO_2_ in a humid atmosphere), 10 μL of MTT (5 mg/mL in phosphate-buffered saline, PBS) was added to each well, and the plate was incubated for a further 3 h (37 °C). After that, the supernatant was discarded and replaced by 100 µL of DMSO to dissolve formazan crystals. The absorbance was then read at 550 nm by spectro-photometry. For all concentrations of compound, cell viability was expressed as the percentage of the ratio between the mean absorbance of treated cells and the mean absorbance of untreated cells. Three independent experiments were performed, and the IC_50_ values (i.e., concentration half inhibiting cell proliferation) were graphically determined by using GraphPad Prism 4 software (2019).

#### 4.2.3. PD-L1, VEGFR-2, and c-Myc Relative Quantification by Flow Cytometry

To study the effect of the compounds on every biological target in cancer cell lines the compounds were used at a 100 or 10 μM dose, depending on their IC_50_ and the treated cell line. Simpler derivatives and BMS-8 were always tested at 100 μM, and sorafenib at 10 μM. Table 7 shows the doses for the rest of tested compound.

For the assay, 10^5^ cells per well were incubated for 24 h with the corresponding dose of the tested compound in a total volume of 500 μL of their growth media.

To detect membrane PD-L1 and VEGFR-2, after the cell treatments, they were collected, fixed with 4% in PBS paraformaldehyde, and stained with FITC Mouse monoclonal Anti-Human VEGFR-2 (ab184903) and Alexa Fluor^®^ 647 Rabbit monoclonal Anti-PD-L1 (ab215251).

For the detection of total PD-L1, VEGFR-2, and c-Myc, the procedure was the same as for membrane targets, but after fixation, a treatment with 0,5% in PBS TritonTM X-100 was performed. The antibodies used for PD-L1 and VEGFR-2 were the ones previously described, and for the detection of c-Myc, it was FITC Rabbit monoclonal anti-c-Myc (ab223913).

#### 4.2.4. Protein Thermal Shift for Studying the Interaction between the Compounds and PD-L1

PTS assay was performed by following the instructions indicated in Protein Thermal ShiftTM Dye Kit (Applied Biosystems reference 4461146), using 2 μL of a 0.1 mg/mL of an aqueous solution of PD-L1 (reference ab167713) and 1.5 μL of a 0.1 mg/mL aqueous solution of the corresponding product. The melting curves were registered by a StepOneTM Real-Time PCR System.

#### 4.2.5. Competitive ELISA Assay

A human PD-1 and human PD-L1 ELISA assay was performed by following the instructions indicated by the manufacturer (Abcam S.A. references ab252360 and ab277712), using 100 pg/mL of both proteins and 100 μM of selected compounds.

#### 4.2.6. ADP Assay

The kinase inhibitory activity of the selected compounds was studied as follows: first 10^5^ HT-29 cells were seeded on a 12 well-plate, and then they were treated with 100 μM of the selected compounds for 24 h. Afterward, cells were collected, and the kinase activity was determined from collected pellets by following the manufacturer’s instructions indicated in the ADP Colorimetric Assay Kit II (Abcam S.A. reference ab282932).

#### 4.2.7. Tube Disruption and Formation on Matrigel Assay

Wells of a IBIDI 15-well μ-plate for angiogenesis were coated with 15 μL of Matrigel^®^ (10 mg/mL, BD Biosciences Europe) at 4 °C. After gelatinization at 37 °C for 30 min, HMEC-1 cells were seeded at 2 × 10^4^ cells/well in 25 μL of culture medium on top of the Matrigel and were incubated for 30 min at 37 °C while they were attached.

For the tube disruption assay (study of antivascular activity), after 24 h, we checked that the capillary networks were formed on wells, and then 100 μM of compounds was added. After 24 h of treatment, the cultures were observed again, and pictures of each well were taken.

For the tube-formation assay (study of antiangiogenic activity), 100 μM of compounds was added 20 min after seeding HMEC-1 on Matrigel. After 24 h of incubation at 37 °C, the cultures were observed to evaluate the potential formation of capillary networks in the presence of the compounds; again, pictures of each well were taken.

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
