# Peer review of "Synthesis and Biological Evaluation of Small Molecules as Potential Anticancer Multitarget Agents"

_ijms, 2022, doi:10.3390/ijms23137049_

Round 1

Reviewer 1 Report

The manuscript describes the preparation and biological evaluation of twenty-six triazole based derivatives designed for targeting both PD-L1 and VEGFR-2. The antiproliferative activity of these molecules on several tumor cell lines (HT-29, A-549 and MCF-7) and on the non-tumor cell line HEK-293 has been determined. The effect on the above mentioned biological targets were evaluated for some selected compounds. Compound 23 bearing a p-chlorophenyl group showed results better than sorafenib in  down-regulation of VEGFR-2 and similar effect to BMS-8 on both PD-L1 and c-Myc proteins. The manuscript is potentially interesting and it could be published after major revision.

The comments are as follows.:

  1. The rational for the development of small molecules bearing a triazole central ring that could target both PD-L1 and VEGFR-2 in tumor cells was not fully explained. I suggest to insert in the introduction section the structures of know molecules characterized by the presence of 1,2,3-triazole ring targeting PD-L1 and/or VEGFR-2. VEGFR inhibitors approved for clinical use should be inserted in the introduction section. As 1,2,3-triazole targeting VEGFR-2 please see: DOI: 10.1016/j.ejmech.2020.113083,
  2. Most of the synthesized compounds have been previously published.
  3. Measurement of potential enzyme inhibitory activity (IC50) was not reported for the most promising candidates demonstrating VEGFR-2 inhibition activity. Also the inhibitory activities of the compounds against the PD-1/PD-L1 interaction should be evaluated
  4. For the most active compound, the chick chorioallantoic membrane (CAM) assay can utilized as an in vivo technique to evaluate the anti-vascular effects

Author Response

The manuscript describes the preparation and biological evaluation of twenty-six triazole based derivatives designed for targeting both PD-L1 and VEGFR-2. The antiproliferative activity of these molecules on several tumor cell lines (HT-29, A-549 and MCF-7) and on the non-tumor cell line HEK-293 has been determined. The effect on the above mentioned biological targets
were evaluated for some selected compounds. Compound 23 bearing a p-chlorophenyl group showed results better than sorafenib in down-regulation of VEGFR-2 and similar effect to BMS8 on both PD-L1 and c-Myc proteins. The manuscript is potentially interesting and it could be published after major revision.
The comments are as follows.:
1. The rational for the development of small molecules bearing a triazole central ring that could target both PD-L1 and VEGFR-2 in tumor cells was not fully explained. I suggest to insert in the introduction section the structures of know molecules characterized by the presence of 1,2,3-triazole ring targeting PD-L1 and/or VEGFR-2. VEGFR inhibitors approved for clinical use should be inserted in the introduction section. As 1,2,3-triazole targeting VEGFR-2 please see: DOI: 10.1016/j.ejmech.2020.113083.
Answer: we have rewritten the introduction and inserted figure 2 with the
structures of clinically approved VEGFR inhibitors. As regards the introduction of the structures of known molecules characterized by the presence of 1,2,3-triazole ring targeting PD-L1 and/or VEGFR-2, we have not considered it appropriate.
Several papers have been published regarding this issue. In these papers many structures are synthesized from a general structure. We do not believe adecuate the insertion of these general structures in our manuscript. Instead we have included references 13-19 dealing with the synthesis and biological evaluation of 1,2,3-triazole ring compounds targeting VEGFR2.
2. Most of the synthesized compounds have been previously published.
Answer: 1-14 structures have been previously synthesized. 15-26 structures are new ones. Our goal was the synthesis of compounds, either previously
synthesized or new ones, that had some common structural features but with different chemical functions in order to obtain conclusions on structure activity relationships.
3. Measurement of potential enzyme inhibitory activity (IC50) was not reported for the most promising candidates demonstrating VEGFR-2 inhibition activity. Also the inhibitory activities of the compounds against the PD-1/PD-L1 interaction should be evaluated.
Answer: We have performed and ADP assay to establish the kinase activity in treated cell cultures (see p.2.3.7 in the revised manuscript) and a competitive ELISA assay to determine the PD-L1/PD-1 blocking activity of the selected compounds.(see p. 2.3.6)
4. For the most active compound, the chick chorioallantoic membrane (CAM) assay can utilized as an in vivo technique to evaluate the anti-vascular effects
Answer: Antivascular as well as antiangiogenic activity of the selected derivatives by tube formation assay on 3D Matrigel HMEC-1 cell cultures have been carried out and have been included in the revised manuscript (see p.2.3.8).

Reviewer 2 Report

The authors synthesized 26 triazole based derivatives and the derivatives were biologically evaluated as multitarget inhibitors of VEGFR-2, PD-L1 and c-Myc proteins. 

The antiproliferative activity of these synthesized molecules on tumor cell lines (HT-29, A-549 and MCF-7) and on the non-tumor cell line (HEK-293) has been determined. The authors demonstrate that compound 23 showed results better than sorafenib in down-regulation of VEGFR-2 and similar effect to BMS-8 on both PD-L1 and c-Myc proteins.

The manuscript can be reconsidered for publication in International Journal of Molecular Science after revision based on comments below. 

  1. The technical background for choosing triazoles should be added in the introduction part.
  2. The definition of mPD-L1 and tPD-L1 is missing. According to table 2, compound 15-17 has effect on PD-L1 of  HT-29, but it is not on PD-L1 of HT-29 based on table 3. What makes the difference ?
  3. The discussion on the result should be correct. For example, on line 187, analogue 21 in HT-29 is not showing any effect on c-Myc. In addition to that, on line 189, compound 18-20 is also showing the effect on c-Myc. 
  4. On line 232, why is the electron donating groups enhancing antiproliferative action and vice versa?
  5. Why is the viability data of compound 24-26 missing in figure 4?
  6. According to table 5, the dTm is changed. In the case of BMS-8, it is +4 but the change by compound 21-23 is - value. Is there any specific reason for that?

  1. Typos should be corrected for example on line 150, 180 and so forth.
  2. The terminology for color should be matched in figure 5. (salmon-orange)
  3. Caption for Figure 1 is missing. And the Figure number should be edited in figure 2.
  4. On line 250, regarding the future work, it is better to move to the end of the manuscript in the conclusion part. 

Author Response

The authors synthesized 26 triazole based derivatives and the derivatives were biologically  evaluated as multitarget inhibitors of VEGFR-2, PD-L1 and c-Myc proteins.
The antiproliferative activity of these synthesized molecules on tumor cell lines (HT-29, A-549 and MCF-7) and on the non-tumor cell line (HEK-293) has been determined. The authors demonstrate that compound 23 showed results better than sorafenib in down-regulation of VEGFR-2 and similar effect to BMS-8 on both PD-L1 and c-Myc proteins.
The manuscript can be reconsidered for publication in International Journal of Molecular Science after revision based on comments below.
1. The technical background for choosing triazoles should be added in the introduction part.
Answer: The introduction has been rewritten and new references related to 1,2,3-triazole ring-containing structures, that have been shown to be active on VEGFR2 inhibition, have been added.
2. The definition of mPD-L1 and tPD-L1 is missing. According to table 2, compound 15-17 has effect on PD-L1 of HT-29, but it is not on PD-L1 of HT-29 based on table 3. What makes the difference ?
Answer: We have added the definitions in the text. According to these, table 2 reflects the results obtained for membrane proteins (PD-L1 and VEGFR-2) These dates were obtained when we measure PD-L1 by flow cytometry without permeabilization of cells. Table 3 shows the results obtained when we measured total PD-L1, that is membrane and cytosolic. In this case, we performed the assay by permeabilizing cells.
The fact is that there are some derivatives that are able to affect the target that is located in the membrane while the inside one remains unaltered. The important thing is that compounds were able to block the binding of PD-L1 cancer cell to PD1 T cell and that interaction occurs in the interface between both kind of cells.
The discussion on the result should be correct. For example, on line 187, analogue 21 in HT-29 is not showing any effect on c-Myc. In addition to that, on line 189, compound 18-20 is also showing the effect on c-Myc.
Answer: We have checked and corrected all the mistakes with the compound numbers.
3. On line 232, why is the electron donating groups enhancing antiproliferative action and vice versa?
Answer: We are not completely sure but it is like as amphitic carácter of the molecule increases, cells became more susceptible to damage.
4. Why is the viability data of compound 24-26 missing in figure 4?
Answer: We have rebuilt these figure in order to reflect all the new assays we have carried out in the process of revising the manuscript. Now this is not Figure 4 but Figure 7 and only chloro derivatives results are shown.
5. According to table 5, the dTm is changed. In the case of BMS-8, it is +4 but the change by compound 21-23 is - value. Is there any specific reason for that?
Answer: According to the fundamentals of this technique, the reason is that an increase of Tm means that the binding of its target leads to a stabilisation of the target. While a decrease in Tm is probably due to the fact that the binding between the ligand and its target leads to destabilisation of the protein possibly that binding between the two leads to destabilisation of the protein.
1. Typos should be corrected for example on line 150, 180 and so forth.
Answer: This has been corrected
2. The terminology for color should be matched in figure 5. (salmon-orange).
Answer: This has been corrected.
3. Caption for Figure 1 is missing. And the Figure number should be edited in figure 2.
Answer: This has been corrected.
4. On line 250, regarding the future work, it is better to move to the end of the manuscript in the conclusion part.
Answer: This has been corrected

Round 2

Reviewer 1 Report

The authors have properly answered to all my questions. The manuscript can be published in the present form. 

Author Response

All is correct according the reviewer

Reviewer 2 Report

The manuscript is revised with more information. It is recommended to add the answer of 1,3 and 5 to the revised manuscript properly.

Author Response

The answers to questions 1,3 ang 5 of reviewer have been added to the manuscript (see manuscript file)